# Polycaprolactone Electrospun Nanofiber Membrane with Sustained Chlorohexidine Release Capability against Oral Pathogens

**DOI:** 10.3390/jfb13040280

**Published:** 2022-12-07

**Authors:** Zi-Jian Chen, Jia-Cheng Lv, Zhi-Guo Wang, Fei-Yu Wang, Ren-Huan Huang, Zi-Li Zheng, Jia-Zhuang Xu, Jing Wang

**Affiliations:** 1Department of Stomatology, Shanghai Tenth People’s Hospital, Tongji University School of Medicine, Shanghai 200072, China; 2State Key Laboratory of Polymer Materials Engineering, College of Polymer Science and Engineering, Sichuan University, Chengdu 610065, China; 3Department of Clinical Cosmetology, Xiangyang No. 1 People’s Hospital, Hubei University of Medicine, Xiangyang 441001, China; 4West China School of Nursing, Sichuan University/West China Hospital, Sichuan University, Chengdu 610041, China; 5State Key Laboratory of Oral Diseases, National Clinical Research Center for Oral Diseases, West China Hospital of Stomatology, Sichuan University, Chengdu 610032, China

**Keywords:** polycaprolactone, electrospinning, oral drug delivery, periodontal dressings

## Abstract

Multiple-pathogen periodontal disease necessitates a local release and concentration of antibacterial medication to control inflammation in a particular location of the mouth cavity. Therefore, it is necessary to effectively load and deliver medicine/antibiotics to treat numerous complex bacterial infections. This study developed chlorhexidine (CHX)/polycaprolactone (PCL) nanofiber membranes with controlled release properties as periodontal dressings to prevent or treat oral disorders. Electrostatic spinning was adopted to endow the nanofiber membranes with a high porosity, hydrophilicity, and CHX loading capability. The release of CHX occurred in a concentration-dependent manner. The CHX/PCL nanofiber membranes exhibited good biocompatibility with human periodontal ligament stem cells, with cell viability over 85% in each group via CCK-8 assay and LIVE/DEAD staining; moreover, the good attachment of the membrane was illustrated by scanning electron microscopy imaging. Through the agar diffusion assay, the nanofiber membranes with only 0.075 wt% CHX exhibited high antibacterial activity against three typical oral infection-causing bacteria: *Porphyromonas gingivalis*, *Enterococcus faecalis*, and *Prevotella intermedia*. The results indicated that the CHX/PCL nanofiber holds great potential as a periodontal dressing for the prevention and treatment periodontal disorders associated with bacteria.

## 1. Introduction

Oral bacteria are dispersed throughout the mouth, and 1 g of plaque may contain up to 100 billion microorganisms [1]. Most of these bacteria, including *Porphyromonas gingivalis* and *Staphylococcus aureus*, are pathogenic. These microorganisms are mostly responsible for the two types of oral disorders, dental caries, and gingival bleeding/periodontitis. Dental caries and periodontitis are not caused by a single causal pathogen, instead by a polymicrobial community that functions synergistically [2,3]. Dental disorders may induce systemic, cardiovascular, and brain diseases if not addressed promptly [4,5]. Therefore, preventing oral antibacterial effects has become a critical global concern.

Broad-spectrum antibacterial drugs, such as penicillin, macrolides, and tetracyclines, are commonly used in dental clinics and have exhibited clinical efficacy. However, the majority of antibiotics are required to be taken orally by the patients [6]. Drug absorption may result in systemic reactions such as allergies, nephritis, hematologic complications, gastrointestinal issues, neurological illnesses, and skin allergy symptoms [7]. To minimize systemic adverse effects and bacterial resistance, local administration and continuous release of antibiotics have been recommended to treat persistent bacterial infections [8]. Periodontal dressings are among the most commonly used oral antibacterial products. They have many benefits, such as wound protection, preventing wound infection, and helping the underlying tissues heal. However, once it is in the mouth, a periodontal dressing may be colonized by bacteria and become infected, which can cause the structure to fail and break. Thus, anti-infection antibiotic-infused periodontal dressings have considerable potential for application in oral bacterial treatments [9,10].

Chlorhexidine (CHX)—a common oral antibiotic—hinders plaque development by altering bacterial cohesiveness, thus preventing germs from adhering to the tooth surface. It kills bacteria by damaging their cell walls via interactions with negative ions [11]. CHX is extensively used in mouthwashes to inhibit bacterial growth and prevent cavity development. However, mouthwashes do not satisfy the local or long-term management requirements. Furthermore, using CHX for an extended period is associated with numerous drawbacks, e.g., the absence of modulatory effects on oral biofilms. Therefore, long-term therapeutic use of CHX is not recommended [12].

To date, utilizing a CHX/polycaprolactone (PCL) membrane as a periodontal dressing to prevent or treat oral disorders has only been rarely done. In this study, the electrospinning method was employed to construct a nanofiber membrane and to demonstrate whether the CHX-loaded dressing can achieve sustained drug release and bactericidal action (Figure 1 summarizes the concept). Overall, this research focused on the assembly of CHX-loaded membranes with different CHX concentrations, as well as the evaluation of the cytotoxicity and antibacterial potential against *Prevotella intermedia*, *Porphyromonas gingivalis*, and *Enterococcus faecalis*.

We present two null hypotheses here: (i) there will be cytotoxicity between various CHX-loaded membranes with increase in CHX concentration; (ii) the antibacterial capability will not significantly differ according to the concentration of CHX.

## 2. Materials and Methods

In this work, the biodegradable PCL pellets (Mv = 6 × 10^4^ g·mol^−1^) were obtained from Shenzhen Guanghua Weiye Industrial Co., Ltd., Shenzhen, China. CHX was purchased from Beijing Solaibao Technology Co., Ltd., Beijing, China. Dichloromethane (DCM) and N, N-dimethylformamide (DMF) were purchased from Chengdu Kelong Chemical Reagent Factory, Chengdu, China. The study was conducted in accordance with the Helsinki Declaration, and the protocol was authorized by the Ethics Committee of Shanghai Tenth People’s Hospital (**ID:22K275**).

### 2.1. Preparation of CHX/PCL Nanofiber Membranes

PCL pellets (1.8 g) were dissolved in 10 mL of a DCM/DMF solution (volume ratio = 80/20). Different amounts of CHX (0.05, 0.075, 0.1, 0.125, and 0.15 wt%) were homogenously mixed with the PCL solution by stirring for 3 h at 25 °C. The mixture was used for electrospinning at an infusion speed of 0.2 mm/min. The working distance between the collector and spinneret was 15 cm, and the voltage was 16 kV; the positive and negative voltages were 13 and −3 kV, respectively. The collection speed of the collector was 100 rpm. Electrospinning was performed at 30 °C and 45% humidity. A series of CHX/PCL nanofiber membranes were obtained after drying at room temperature for 1 h.

### 2.2. Characterization

Scanning electron microscopy (SEM)

The morphologies of the CHX/PCL nanofiber membranes were examined via SEM (Nova NanoSEM450, FEI, Hillsboro, OR, USA) at an accelerating voltage of 5 kV after sputtering a thin gold layer. Image analysis software (ImageJ) was used to measure the diameters of the nanofibers.

Rheological Test

The viscosity of the spinning fluid with different CHX concentrations was measured by a rotary rheometer (MCR 302, Anton Paar GmbH, Graz, Austria). Three microliters of spinning fluid were placed between the lamina, which had a diameter of 50 mm and an angle of 1°. The shear rate ranged from 0.01 to 100 s^−1^ with five points between each order of magnitude measured at 25 °C.

Water contact angle (WCA)

The hydrophilicity of the CHX/PCL nanofiber membranes was qualitatively assessed by measuring WCA using a DSA25 system (Krüss, Hamburg, GER). Four microliters of deionized water were dropped onto the membrane, and the static contact angle was recorded using image analysis software. For each sample, the average value of five measurements was calculated.

Fourier-transform infrared (FTIR) spectroscopy

Functional groups of the CHX/PCL nanofiber membranes were analyzed using an infrared spectrometer (Nicolet 6700, Thermal Scientific, Waltham, MA, USA) equipped with attenuated total reflectance. FTIR spectra were obtained in the wavenumber range of 4000−600 cm^−1^ by averaging 16 scans at a resolution of 4 cm^−1^.

Thermogravimetric analysis (TGA)

Thermal stability of CHX/PCL nanofiber membranes (~5 mg, *n* = 3) was measured via TGA (TG209F1 Iris, NETZSCH, Selb, Germany) at a heating rate of 10 °C/min from 40 to 760 °C under N_2_ atmosphere.

Physical parameter measurements

The nanofiber membranes were cut into rectangular slices (2 × 7 cm^2^). The sample was uniaxially stretched at room temperature using an Instron universal testing instrument (model 5967, Instron Instruments, Norwood, MA, USA). The stress–strain curves of samples at room temperature were obtained.

Release assessment in vitro

The nanofiber membranes with different CHX concentrations were cut into rectangular sheets and weighed. Membranes of similar quality were placed in a centrifuge tube with 3 mL of a phosphate-buffered saline (PBS) solution, which was placed in a shaker for shaking (100 rpm/min). Samples were collected at different release times, and the release of CHX in PBS was measured using an UV spectrophotometer (Lambda 705S, PerkinElmer, Waltham, MA, USA) at 254 nm.

### 2.3. Cytotoxicity Evaluation In Vitro

Human periodontal ligament stem cells (hPDLSCs) were used to evaluate the cytotoxicity of the CHX/PCL electrospun membranes. The hPDLSCs were obtained from premolars, which were selected from patients aged 12–16 years with no periodontitis or gingivitis at the Shanghai Tenth Hospital Department of Orthodontics, and periodontal ligament tissue was isolated from the middle third of the root surface and cultured in a complete medium (α-MEM, Gibco, Invitrogen Corp., Paisley, Scotland, 10% FBS Gibco, Invitrogen Corp., Paisley, Scotland, 1% penicillin-streptomycin, Sangon Biotech (Shanghai) Co., Ltd., Shanghai, China). The P0 cell suspension was then seeded at a concentration of 500 cells/mL in 96-well plates for single cell-derived colony selection. These cells were passaged and expanded for identification and subsequent experiments. Cells from passages three to five were used. Additionally, the different groups of CHX/PCL electrospun nanofiber membranes were cut into 12 mm disc-shaped slices and sterilized under UV light for 24 h (on both sides). The cell viability and proliferation were evaluated using CCK-8. Briefly, the cells were seeded in a 24-well culture plate overnight in the aforementioned medium and then co-cultured with CHX/PCL membranes at different concentrations for 24 and 72 h. In accordance with the manufacturer’s protocol, 10% CCK-8 reagent was added to each well, followed by incubation for 1.5 h, and then the optical density (OD) was measured using a SpectraMax i5 microplate reader (Molecular Devices, Sunnyvale, CA, USA) at a wavelength of 450 nm. The following equation was used to calculate the cell viability of the samples according to the average OD values:$$\mathrm{Cell}\;\mathrm{viability}\left(\%\right)=\frac{\mathrm{OD}\;\mathrm{values}\;\left(\mathrm{sample}\right)}{\mathrm{OD}\;\mathrm{values}\;\left(\mathrm{control}\right)}\times 100\%.$$

After 72 h of co-incubation, an Invitrogen LIVE/DEAD Viability/Cytotoxicity Kit (Thermo Fisher Scientific GmbH, Dreieich, Germany) was used. Briefly, after removing each group of electrospun membranes, the cells were washed thrice with PBS. In accordance with the manufacturer’s protocol, 200 µL of calcein AM and ethidium homodimer-1 dye were added, followed by incubation for 30 min in the dark. Finally, LIVE/DEAD staining was analyzed using a fluorescence microscope (Nikon Corporation, Tokyo, Japan). To observe the cells adhered to the surface of the CHX/PCL membrane, the SEM images of cell-seeded samples were taken after 72 h of co-culturing.

### 2.4. Antibacterial Activity

The antibacterial properties of the CHX/PCL electrospun membranes were tested via growth inhibition of *P. gingivalis* (ATCC33277), *E. faecalis* (ATCC 19433), and *P. intermedia* (ATCC 25611) using both agar diffusion and broth transfer assays. A BHI broth solution was prepared, which contained 21 g of BHI broth (Oxoid Limited, Basingstoke, UK) with 5% yeast powder (Sangon Biotech (Shanghai) Co., Ltd., Shanghai, China) in 1 L of ddH_2_O. To prepare BHI agar plates, agar was added to the BHI broth base solution at 15 g/L, and Columbia Blood Agar plates were used under anaerobic conditions (AnaeroPack, Mitsubishi Gas Chemical Company, Tokyo, Japan) during culturing of *P. gingivalis* and *P. intermedia*. The bacterial suspension in the BHI broth solution was cultured at 37 °C with shaking at 180 rpm. The densities of the suspensions were measured using a SpectraMax i5 microplate reader (Molecular Devices, Sunnyvale, CA, USA) at a wavelength of 600 nm. They were adjusted to OD600 = 0.5, which corresponded to 1 × 10^7^ cells/Ml, and the fibers were prepared as described previously. Bacterial suspensions in BHI broth with OD600 = 0.5 were separated into different groups with 50 µL of suspension in 950 µL of BHI broth. Each of the CHX/PCL membranes was immersed in the bacterial suspensions (1 Ml) and cultured overnight at 37 °C and 180 rpm (*P. gingivalis* and *P. intermedia* groups were incubated under anaerobic conditions, as described previously). The optical absorption was measured at 600 nm (Molecular Devices, Sunnyvale, CA, USA). Different groups of bacterial suspensions were diluted 10^6^ times with sterilized PBS (Sigma-Aldrich Corp., St. Louis, MO, USA). Next, 100 µL of the diluted bacterial solution was spread on BHI agar plates or Columbia Blood agar plates under anaerobic conditions, and all the plates were incubated at 37 °C for 24 h.

## 3. Results

### 3.1. Morphology and Chemical Compositions of CHX/PCL Nanofiber Membranes

Figure 2 shows the morphology of the electrospun CHX/PCL membranes. The CHX/PCL electrospun nanofibers exhibited a smooth surface and random alignment (Figure 2a–f). The fibers of the pure PCL nanofibers had a wide diameter range, with an average diameter of 454 nm (Figure 2g). With the addition of CHX, the diameters of the electrospun nanofibers decreased (Figure 2g–i). It is speculated that the surfactant CHX reduced the viscosity of the spinning solution. Both the diameter and size decreased with an increase in CHX content (Figure 2). For instance, the diameter of the 0.075 wt% CHX/PCL nanofibers ranged from 100 to 400 nm, with an average value of 274 nm, and the diameter of the 0.15 wt% CHX/PCL fibers ranged from 50 to 350 nm, with an average value of 191 nm.

Figure 3a shows the viscosity variation with shear stress of samples with different CHX concentrations. It was found that at the same shear rate, the viscosity of the spinning solution gradually decreased with the increase in CHX concentration. This is primarily because CHX has a surfactant effect, which reduces the viscosity of the spinning solution. This phenomenon also explains the observation that the diameter of spun fiber decreased with increased CHX concentration, with the other electrospinning parameters remaining the same.

Figure 3b compares the FTIR spectra of pure PCL and 0.15 wt% CHX/PCL. The FTIR spectrum of the CHX/PCL nanofiber film was consistent with that of the pure PCL membrane. This is because the CHX content was below the limit of detection. The absorption peaks at 2940 and 2860 cm^−1^ were mainly caused by the stretching of the −CH_2_− bond, and carbon-based absorbance appeared at 1720 cm^−1^. To investigate the existence of CHX in the fiber membranes, we extracted the CHX/PCL membrane and analyzed the extract using FTIR spectroscopy. The results are shown in Figure 3c. The absorption peak at 1600 cm^−1^ is attributed to the aromatic ring C=C, and the vibration of the C=N group formed a moderate rise in the region of 1690–1590 cm^−1^. In the region of 800–700 cm^−1^, the peak was formed by benzene ring para-transposition, which confirmed the presence of CHX on the fiber membrane.

The mechanical properties of the material are shown in Figure 3d. With the increase in CHX content, the maximum tensile stress and elongation at break of the fiber film decreased. The elongation at break decreased to about 40% at 0.1 wt% and then had little change with further increased CHX content. The main reason is that with the increase in CHX concentration, the fiber diameter becomes smaller, which affects the mechanical properties of the material.

The thermal stability of the CHX/PCL nanofiber membrane was evaluated using TGA, as shown in Figure 3e. The electrospun fibers did not lose weight in the temperature range of 104–108 °C, indicating that the solvent had entirely evaporated during the spinning. Compared with the pure PCL fibers, the incorporation of CHX did not affect the thermal stability. For example, the initial decomposition temperature of the 0.125 wt% CHX/PCL membrane was 390.2 °C, and the maximum cracking temperature was 412.9 °C.

The wettability of the CHX/PCL nanofiber membranes was evaluated via WCA measurements. As shown in Figure 3f, the WCA of the pure PCL nanofiber membranes was 144.8°, indicating obvious hydrophobicity. With an increase in the CHX content, the WCA of the CHX/PCL nanofiber membranes decreased. The WCA of the 0.15 wt% CHX/PCL nanofiber membranes was 49.7°. The WCA of the CHX/PCL membranes suddenly decreased when the CHX content reached 0.125 wt%; thus, the CHX content reached a critical value, at which the membranes changed from hydrophobic to hydrophilic.

### 3.2. Release Assessment In Vitro

The in vitro release of CHX from CHX-containing nanofiber membranes is shown in Figure 4a. CHX had a fast release rate in the first 480 min, which then decreased. The experimental groups with 0.05, 0.075, 0.1, 0.125, and 0.15 wt% CHX supplementation released 15, 19, 23.5, 25, and 26 µg/Ml in the first 480 min, respectively. After 5760 min, the release rate of CHX gradually decreased until the fifth day. The amount of CHX released did not increase exponentially when the amount of CHX in the fiber increased from 0.05 to 0.15 wt%, possibly owing to the fact that most of the CHX released was from the surface of the fiber while the CHX inside the fiber was packed into the fiber. As shown in Figure 4b, at 7200 min, the release percentage of the 0.05 wt% CHX content group was the highest, reaching 81%, while the release percentage of the 0.15 wt% CHX content group was the lowest (only about 69%) and gradually decreased with the increased CHX concentration.

### 3.3. Cytotoxicity Evaluation In Vitro

The cell viability for different concentrations of CHX/PCL membranes was evaluated using the CCK-8 assay. According to the OD value, with an increase in the CHX concentration (0.05, 0.075, 0.1, 0.125, and 0.15 wt% CHX/PCL), the cell viability was calculated using the absorption value of each group after 24 and 72 h of co-culturing. As shown in Figure 5a, there were no significant differences between the control and CHX/PCL groups. However, compared with the control group, the high-CHX concentration groups (0.125 and 0.15 wt%) were statistically significant (* *p* < 0.05 and ** *p* < 0.005, respectively, at 24 h and * *p* < 0.05 for both at 72 h). Meanwhile, Figure 5b indicates the high cell viability of each group at 24 and 72 h, which suggests low cytotoxicity for each concentration of CHX electrospun membranes. After 72 h, the 0.05 wt% CHX/PCL fiber membrane exhibited statistically significant differences compared with the 0.125 wt% CHX/PCL (* *p* < 0.05) and 0.15 wt% CHX/PCL (*** *p* < 0.001) membranes, which proves that compared with low concentrations, high concentrations of CHX resulted in lower cell viability. Nevertheless, the cell viability was >85% for all the groups, indicating that hPDLSCs continuously proliferated and grew for 72 h in the presence of CHX. For further evaluation of the biocompatibility, the fluorescence microscope image of the LIVE/DEAD assay was examined, as shown in Figure 5c; there were marginal differences in cell proliferation among different CHX concentrations. After 72 h of co-culturing, live cells (shown as green) occupied most of the area for each group, and dead cells (shown as red) were rarely observed. Moreover, compared with that in the control group, the density of viable cells was slightly reduced for 0.15 wt% CHX, which was consistent with the results of the CCK-8 assay. In conclusion, the CHX/PCL fiber membranes exhibited high biocompatibility. Thus, the LIVE/DEAD assay suggests acceptable biocompatibility of the CHX/PCL fiber membrane. To verify the cell adhesion, we performed SEM after 72 h. SEM images of the cells that adhered to the surface of the fiber membranes are shown in Figure 5d. As expected, hPDLSCs were seeded on the surface for each group. Additionally, a denser layer of cells formed and spread over the surface. According to these observations, different concentrations of CHX/PCL fiber membranes are suitable for cell attachment and proliferation, which means that the membranes are more likely to be utilized in the oral cavity.

### 3.4. Antibacterial Activity

The antibacterial properties of different concentrations of CHX/PCL fiber membranes against three typical oral pathogens—*P. gingivalis*, *E. faecalis,* and *P. intermedia*—were determined according to the OD600 value and an agar diffusion assay. Different 12 mm disc-shaped samples were immersed in the bacterial suspensions in the property condition overnight. Following Figure 6a, the absorption value shows that each group had a statistically significant difference compared with the control group (**** *p* < 0.001). In particular, for the high-concentration CHX/PCL groups, the OD values approached the BHI medium, indicating that the bacterial cells were unable to clone during the initial time. The results of agar diffusion are shown in Figure 6b. As shown, the high 0.1, 0.125, and 0.15 wt% fiber membranes resulted in inhibition against both oral gram-negative (*P. intermedia, P. gingivalis*) and gram-positive bacteria (*E. faecalis*). The results also indicated that the CHX concentration increment from 0.1 to 0.15 wt% led to higher antibacterial activity against both gram-negative and gram-positive bacteria, as confirmed by the smaller number of colony-forming units.

## 4. Discussion

In this study, periodontal dressings were found to be at risk of reinfection [13]. Once the periodontal dressing is exposed to the oral cavity, it is susceptible to bacterial infection, which may cause rapid structural rupture. Recent research has indicated that periodontal dressings including natural (chitosan, silk fiber, etc.) or synthetic polymers (polycaprolactone (PCL), poly L-lactic acid, etc.) [14,15,16,17,18] contain various antibacterial and antifungal components. Sigusch et al. introduced a periodontal dressing to treat aggressive periodontitis [19]. Compared with the control group, the periodontal dressing group exhibited a substantial reduction in pocket depth. CHX is a commonly used organic antibiotic in dentistry with dose-dependent action against periodontitis- and peri-implant-associated microbes [20,21]. To evaluate the cytotoxic and antibacterial activities of three common oral pathogens in vitro, we used electrospun PCL membranes with different CHX concentrations. Preliminary evidence of their potential therapeutic utility as periodontal dressings was obtained.

Because of their porous nature and large surface area [22], electrospun fibers have considerable potential for incorporating medications and controlling their release in the oral cavity. The morphologies of the CHX-containing fibers were uniform and excellent (Figure 2). There were no discernible changes between the CHX-containing and control (CHX-free) fibers, except for a minor reduction in the fiber diameter in the group with the highest concentration. This was primarily due to the addition of CHX as a surfactant to the spinning solution, which reduced the viscosity of the solution. Therefore, electrospun fibers produced by electrostatic spinning have comparatively small diameters. Electrospun fibers composed of doped CHX exhibited sporadic fractures and adhesions; however, the overall fiber shape was satisfactory. The phenomenon of fiber diameter reduction after CHX doping was indicated by the WCA. This was ascribed to the fact that CHX contains abundant hydrophilic amino groups. The hydrophilicity of the material is not only related to the existence of hydrophilic and hydrophobic groups on the surface of the material, but is also to the microstructure of the material surface [23]. As shown in Figure 2, the fiber diameter statistics show that with the increase in CHX concentration, the fiber diameter became smaller. Therefore, the surface of the fiber membrane prepared by electrospinning is relatively dense, and the air resistance in the pores is small. At the same time, as a water-soluble molecule, the increase in CHX concentration on the nanofibers will cause osmotic pressure, and water permeates the pores; thus, WCA decreases. Because of the rich porous structure of the CHX-modified electrospun membrane, the fiber diameter was small, resulting in relatively compact surface, low air resistance in the pores, and reduction in the contact resistance. Electrospun membranes have hydrophilic mechanical characteristics. The hydrophilicity of a material may provide excellent interaction with water, facilitating the release of pharmaceuticals. Owing to the more fragile fibers of the electrospun membrane, the doping of CHX reduced the elastic modulus of the electrospun membrane. Figure 3 presents the results of an infrared examination, which indicated that a modest quantity of CHX did not significantly alter the distinctive peak of pure PCL. The injection of CHX may not affect the chemical characteristics of the PCL electrospun membrane. Infrared characterization of an electrospun membrane doped with CHX was also performed. These data indicated that CHX was successfully introduced. The drug release process in the fiber is primarily divided into two stages. The first stage is the release of the drug attached to the fiber surface, which is a rapid drug release process; the second is the release of the drug covered by the fiber, and the drug release rate in this process primarily depends on the degradation process of the fiber material PCL [24]. As shown in Figure 4, all the CHX-doped electrospinning films showed the highest drug release rate and quantity in the first 480 min, which was because of the rapid release of CHX attached to the fiber surface. Subsequently, the release rate decreased steadily from 480 min to 7200 min. This slow release of CHX was caused by PCL electrospinning fibers that have an encapsulation effect on CHX, and the encapsulation effect increased with increased addition of CHX. Although different CHX/PCL films with different concentrations have different degrees of CHX encapsulation, it does not affect the release of sufficient CHX from the film to play a bactericidal role. From 1440 to 7200 min, the release was gradual, and the release rate was constant. During this period, the release rates of the electrospun membranes with different CHX concentrations were comparable, which may be explained by their similar shapes. This also indicated that the electrostatically spun membrane has a prolonged influence on drug release. At 4320 min, the percentage release for 0.05 wt% CHX reached 80% of the total content and it remained steady after 4320 min.

Before each proposed application, evaluation of cytotoxicity is necessary. In this work, we used cell viability and cell attachment assays to determine the cytotoxicity of CHX/PCL membranes. A recent study discovered that PCL is a type of harmless, acceptable synthesized substance without negative impacts [25,26]. The absorption values obtained from the CCK-8 assay are shown in Figure 5a, suggesting that none of the CHX concentrations exhibited a substantial toxic effect that compromised the viability of hPDLSCs in vitro. Additionally, the cell viability was determined, as shown in Figure 5b. The high CHX concentration groups (0.125 and 0.15 wt%) had lower values than the other groups, but all groups had values of >85%. According to Karpiski et al. [27], a high CHX concentration may negatively affect cell growth, which is determined by the pH of the surrounding environment. The slow release of nanofibers may counterbalance this effect because of the dose-dependent nature of the effect of CHX on cells. The LIVE/DEAD staining results shown in Figure 5c are comparable to previous findings. In addition, the cell proliferation and adhesion on the surface of the nanofibers indicated biocompatibility [28], as shown in Figure 5d. In the early phase of contact, each group exhibited satisfactory attachment and adhesion, indicating the biocompatibility of the electrospun PCL nanofibers. In conclusion, no detrimental effect on hPDLSCs was observed in vitro.

It has been well established in the literature that when homeostasis is interrupted, the dysbiotic increase in commensal bacteria leads to non-symbiosis of the microbial population, which functions as pathobionts that accelerate the development of disease [1,4,29]. Therefore, the second key purpose of this study was to demonstrate the antibacterial efficacy of CHX/PCL nanofiber membranes against oral infections by employing three common oral pathogens: *P. intermedia* (gram-negative, anaerobic), *P. gingivalis* (gram-negative, anaerobic), and *E. faecalis* (gram-positive, facultative anaerobic). It is well recognized that *P. intermedia* and *P. gingivalis* are closely related in dental biofilms and that their co-colonization is associated with chronic periodontitis [30]. Additionally, refractory periapical periodontitis or persistent root canal infections may be associated with *E. faecalis* [31,32]. According to the OD600 value and the agar diffusion experiments shown in Figure 5a,b, larger quantities of CHX were associated with a stronger inhibitory capability. Specifically, the 0.125 and 0.15 wt% groups inhibited bacterial proliferation in the early phases. The CHX released by the nanofiber membrane is responsible for its antibacterial effect. This may be because of the effective antibacterial activity of CHX, as shown by Xu et al. [33] who reported that the concentration of CHX increased from 50 to 250 μg/mL in an injectable hydrogel combining CHX and nanohydroxyapatite against gram-positive bacteria (*E. faecalis*). Our findings agree with those of Boaro et al. [34], who reported that their composites containing CHX inhibited bacterial growth, except for *P. gingivalis*, for which a low concentration of CHX was required. According to the results of the present study, PCL nanofibers containing CHX have considerable potential for controlling or reducing the activity of *P. intermedia*, *P. gingivalis*, and *E. faecalis*. The antimicrobial activity of these nanofibers suggests that they have a wide range of action against oral species. This action may inhibit the growth and colonization of bacteria in vitro.

Nevertheless, although this study did not investigate the impact on cell apoptosis and cell cycle to provide evidence on the biocompatibility of CHX/PCL membrane, we used the methodologies mentioned above for biomaterial assessment in vitro; future work on biocompatibility evaluation should include multiple methods to provide more powerful evidence. Moreover, because it is difficult to imitate the complex multi-pathogen environment in the oral setting using isolated bacterial species, and because the membrane was not evaluated in vivo, the high antibacterial effect observed here may not directly transfer to an equivalent clinical effect. Thus, more focus should be placed on the imitation of complicated infections and more complete in vivo testing. Despite these limitations, we present more efficient construction of CHX nanofiber membranes by electrospinning and proof of employing the CHX/PCL membranes in periodontal dressing. The null hypotheses may be rejected based on the findings of our investigation.

## 5. Conclusions

Electrostatic spinning was effective for producing CHX/PCL nanofiber membranes. Examination of the relative physicochemical parameters revealed that the CHX/PCL membranes had satisfactory surface shapes. Moreover, with regard to the degradation behavior, tensile strength, contact angle, and biocompatibility, the manufactured membranes exhibited acceptable properties in vitro. In addition, the CHX/PCL nanofiber membranes exhibited high antibacterial activity and antibacterial efficacy against *P. intermedia*, *P. gingivalis*, and *E. faecalis* in vitro. These membranes were not significantly cytotoxic to hPDLSCs. The findings of this study indicate that the method used to prepare CHX-doped PCL nanofiber membranes can be applied in a manner that delivers a sustained antibacterial effect when used as a periodontal dressing.

## Figures and Tables

**Figure 1 jfb-13-00280-f001:**
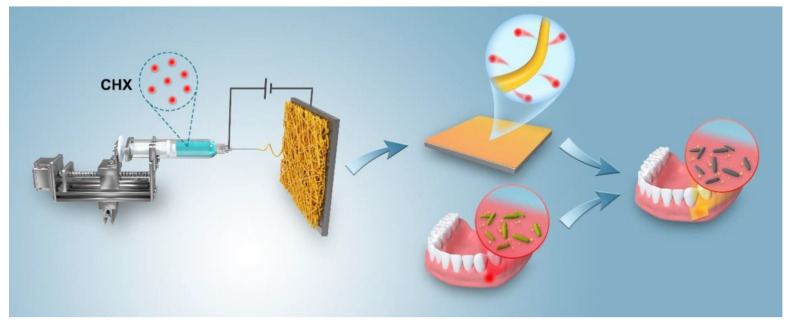
Schematic of the manufacture and application of antibacterial periodontal dressings for treating oral periodontal bacterial infections.

**Figure 2 jfb-13-00280-f002:**
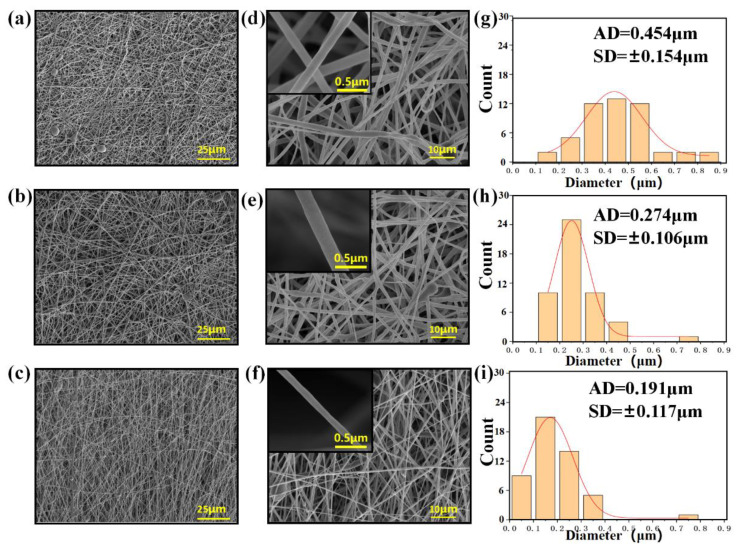
SEM images and diameter distributions of the PCL nanofiber membranes with CHX loading of (**a**,**d**,**g**) 0 wt%, (**b**,**e**,**h**) 0.075 wt%, and (**c**,**f**,**i**) 0.15 wt%.

**Figure 3 jfb-13-00280-f003:**
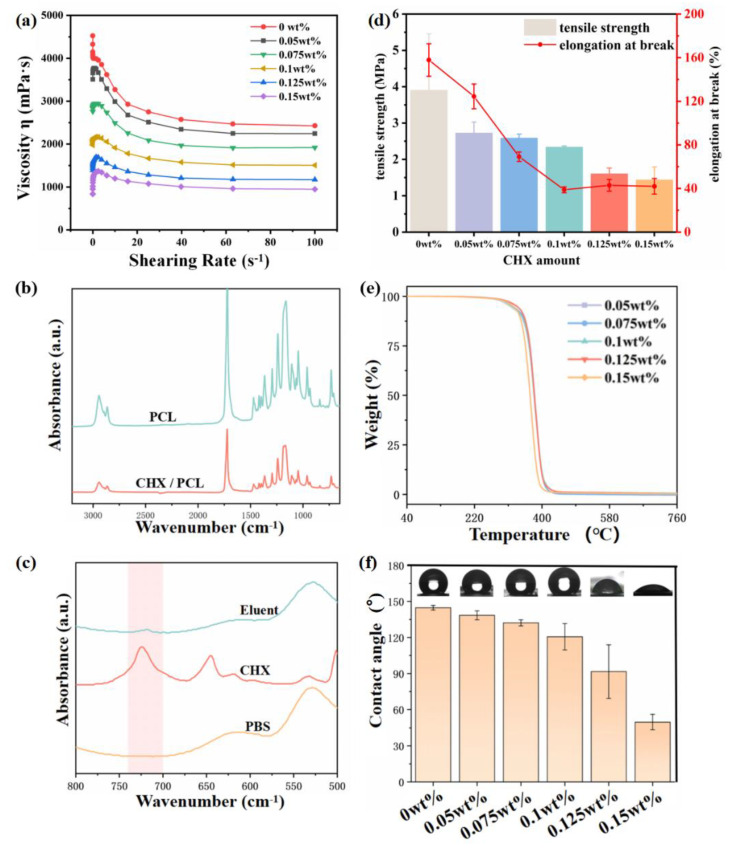
(**a**) Diagram of viscosity variation with shear stress of spinning solution with different CHX concentration; (**b**) FTIR spectra of the 0.15 wt% CHX/PCL and pure PCL nanofiber membranes; (**c**) FTIR spectra of Eluent, CHX, and PBS; (**d**) Maximum tensile stress and elongation at break for samples with different CHX contents; (**e**) TGA curves of CHX/PCL and pure PCL nanofiber membranes; (**f**) WCAs of CHX/PCL and pure PCL nanofiber membranes.

**Figure 4 jfb-13-00280-f004:**
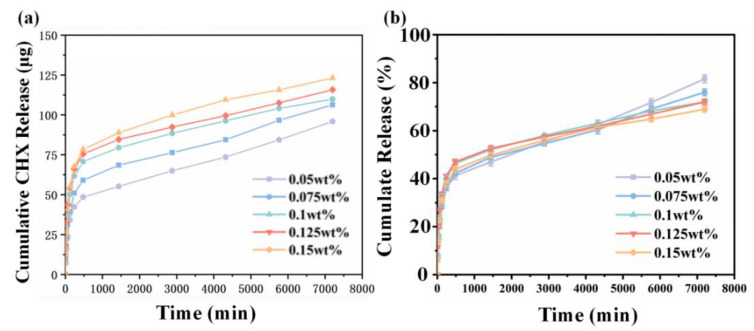
(**a**) Cumulative release and (**b**) percentage release of CHX for CHX/PCL nanofiber membranes with different CHX concentrations in PBS from 0 to 7200 min.

**Figure 5 jfb-13-00280-f005:**
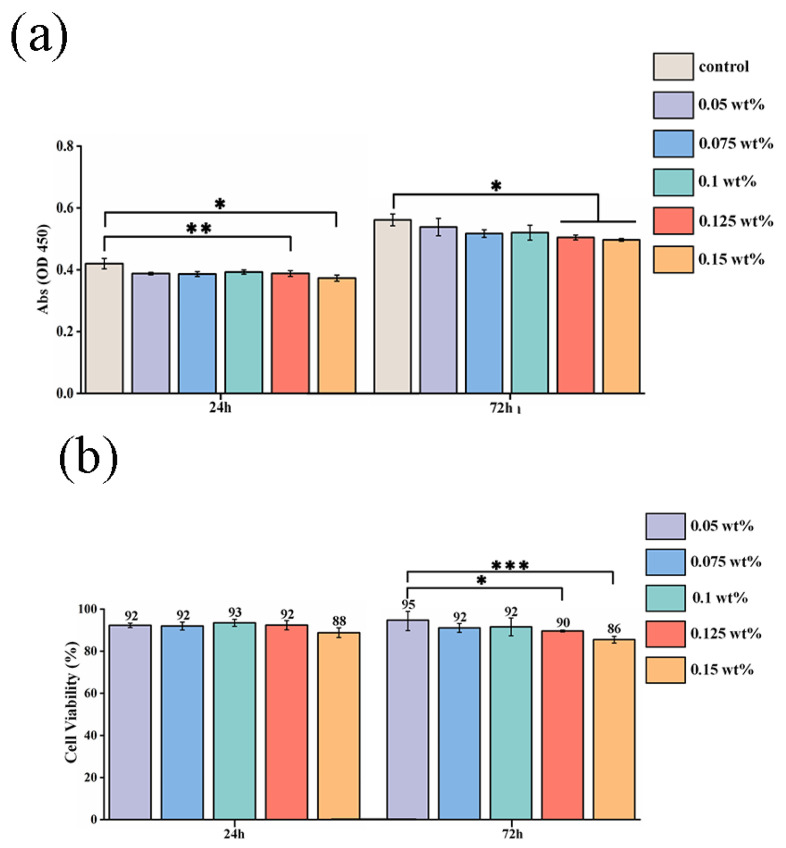
Cytotoxic evaluation of hPDLSCs on CHX/PCL nanofibers: (**a**) CCK-8 assay based on the OD450 value; (**b**) cell viability based on the CCK-8 assay; (**c**) LIVE/DEAD staining images of hPDLSCs; (**d**) SEM image of hPDLSCs on the nanofibers (scale bar = 200 μm & 50 μm) (* *p* < 0.05, ** *p* < 0.01, *** *p* < 0.001).

**Figure 6 jfb-13-00280-f006:**
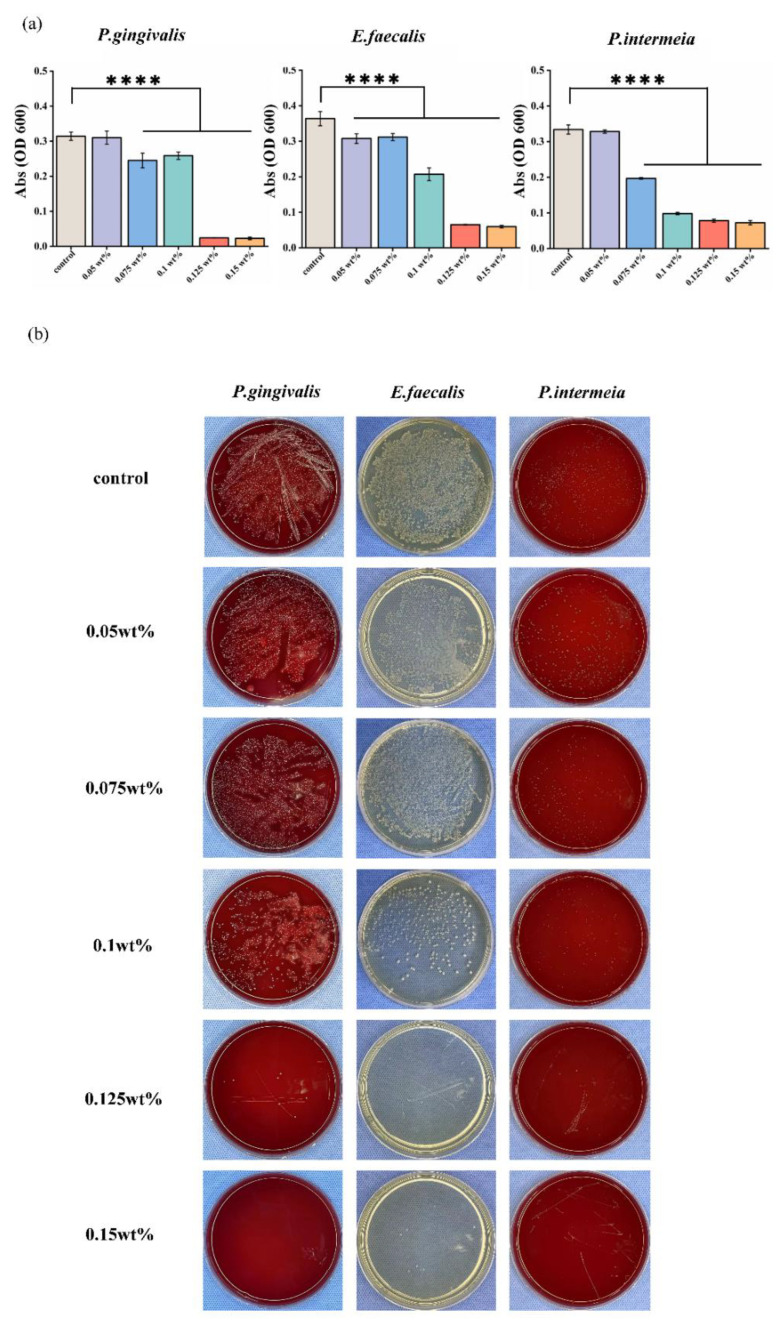
In vitro antibacterial activity of CHX/PCL nanofibers: (**a**) OD600 values of *P. gingivalis*, *E. faecalis*, and *P. intermedia*; (**b**) photographs of agar diffusion results obtained by co-culturing *P. gingivalis*, *E. faecalis*, and *P. intermedia* with different samples (**** *p* < 0.001).

## Data Availability

Not applicable.

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
