# Peer review of "Polycaprolactone Electrospun Nanofiber Membrane with Sustained Chlorohexidine Release Capability against Oral Pathogens"

_jfb, 2022, doi:10.3390/jfb13040280_

Round 1

Reviewer 1 Report

Comments to the Author

The author prepared a kind of antibacterial nanofibrous structure with electrospinning method. The structure and the results were well explored, while some points need to be addressed before publication.

- Line 179, it is mentioned that “The CHX/PCL electrospun nanofibers exhibited a smooth surface and random alignment (Figures 2a–f).”, but the figures do not show the surface morphology. It is suggested to present SEM images with higher magnifications.

- Line 182, “It is speculated that the surfactant CHX reduced the viscosity of the spinning solution.”, Please provide the experimental results for viscosity measurements.

- It is strongly suggested to use the relevant references in the discussion part to confirm the results and explanations.

- Lines 389-391, “meanwhile, with regard to the degradation behavior, tensile strength, contact angle, and biocompatibility, the manufactured membranes exhibited acceptable properties in vitro”. The results for mechanical properties are not presented in the text.

Author Response

Thank you for your comments concerning our manuscript. Those comments are all valuable and very helpful for revising and improving our paper, as well as the important guiding significance to our researches. We have studied these comments and made corrections carefully.

Reviewer 2 Report

Interesting and well-structured study on the biological properties of nanofiber membranes with the addition of chlorhexidine.

Overall work well done with some criticisms listed below:

-In the abstract section insert, in the results part, some numerical references, even if general

- check that all keywords are pubmed MESH terms

-line 38 indicate which oral disorders

-always in the abstract section the techniques used (only the name) must be entered

-In the introduction section, before talking about traditional antimicrobial substances, it is necessary to insert some references regarding the new bioactive materials which, thanks to the release of pharmacologically active active substances, could guarantee the same effect. In this regard, I recommend that you insert the following scientific work in the reference section, which could be of help to the reader:

Lardani L, Derchi G, Marchio V, Carli E. One-Year Clinical Performance of Activa ™ Bioactive-Restorative Composite in Primary Molars. Children (Basel). 2022; 9 (3): 433. Published 2022 Mar 19. doi: 10.3390 / children9030433

-Line 65 and following: this part must be moved to materials and methods such as, for example, sample preparation and removed from the introduction

- at the end of the introduction section the purpose of the study and the null hypotheses of the same must be inserted which will be refuted at the end of the results obtained

-How come those concentrations of antimicrobial agent were selected? What is the rational?

- line 125 remove "using a common method"

- the reference ethics committee for the authorization of the study is missing

- why were 24 and 72 hours selected? Why weren't longer times selected?

-The 5th figure is too small; put it as a separate figure or enlarge it considerably

- Line 346 the bibliographic reference is missing

-As regards the evaluation of cytotoxicity, some considerations on the possible effect of the substance on apoptosis and cell cycle, and not only on cell viability, should be included in the discussion section.

In this regard, I recommend that you insert the following scientific work in the reference section, which could be of help to the reader:

Pagano S, Lombardo G, Costanzi E, et al. Morpho-functional effects of different universal dental adhesives on human gingival fibroblasts: an in vitro study. Odontology. 2021; 109 (2): 524-539. doi: 10.1007 / s10266-020-00569-x

-A section on study limits is missing

Author Response

Thank you very much your useful comments on our manuscript and quality of our manuscript was highly elevated after taking your advice. 

Reviewer 3 Report

In this manuscript authors reported the preparation of controlled Chlorohexidine (CHX) releasing electospun Polycaprolacone (PCL) membranes to be used in periodontal disease, their cytotoxicity with Human periodontal ligament stem cells (hPDLSCs) and antibacterial properties against P. gingivalis, E. faecalis, and P. intermeia were evaluated in vitro. The novelty of the study was found to be below average, since the rationale of the study was not emphasized by comparing the various articles (ie. https://doi.org/10.1016/j.msec.2019.109798) reporting similar controlled drug releasing electrospun PCL periodontal dressing membranes. The manuscript is reporting the results of the standard experiments for such material.

Some points to be addressed by the authors are as follows:

-        - The origins of the materials used are not given.

-       -  The properties of the PCL used for electropspinning is not provided

-      - There is no need to present two different graphs in Figure 4. It is better to present the % of released CHX in the second graph.

-     - The comments on wettability shall be reconsidered. The decrease in the water contact angle is associated directly on the amino groups of CHX only. Since it is a nanofibrous membrane, it can definitely be told that WCA is dependent to surface morphology and porosity of the membrane. As CHX is a water soluble (50% w/v) molecule, presence of CHX with increasing concentration on the nanofibers induce the osmotic pressure and water penetrates through the pores and the WCA decreases.

-    - A phase separation occurred during electrospinning and CHX is mainly located at the nanofiber surface which was released rapidly.

-    - The size of the images in Figure 5 is very low that nothing can easily be evaluated.

-        - Similarly Figure5 can be enhanced.

-      - “CHX contents were comparable, which may explain their similar shapes.” at line 342, what is intended with shape?

-     -  Discussion on release must be enhanced.

- “The slow release of nanofibers” is not a suitable statement, CHX was released.

Author Response

Thank you very much for your comments on our work, they were very constructive and we responded point by point to your comments and used them to revise our manuscript.

Round 2

Reviewer 3 Report

In this revised manuscript authors reported the preparation of controlled Chlorohexidine (CHX) releasing electospun Polycaprolacone (PCL) membranes to be used in periodontal disease, their cytotoxicity with Human periodontal ligament stem cells (hPDLSCs) and antibacterial properties against P. gingivalis, E. faecalis, and P. intermeia were evaluated in vitro. The revisions made in the article enhanced the article.

Some points to be addressed by the authors are as follows:

-        “The properties of the PCL used for electropspinning is not provided” sentence was about the Mn, Mw and/or Mv of the PCL used, Mv of the PCL have been provided in the revised manuscript.

The revised article should be checked for typos

Author Response

We thank your insightful and informed comments, as well as the effort you put into this paper. By employing "Editage," we have improved the writing style and double-checked the whole manuscript for typos. After carefully considering your comment, I have provided a response; please see the attached.
